# L-Tetrolet Pattern-Based Sleep Stage Classification Model Using Balanced EEG Datasets

**DOI:** 10.3390/diagnostics12102510

**Published:** 2022-10-16

**Authors:** Prabal Datta Barua, Ilknur Tuncer, Emrah Aydemir, Oliver Faust, Subrata Chakraborty, Vinithasree Subbhuraam, Turker Tuncer, Sengul Dogan, U. Rajendra Acharya

**Affiliations:** 1School of Management & Enterprise, University of Southern Queensland, Darling Heights, QLD 4350, Australia; 2Faculty of Engineering and Information Technology, University of Technology Sydney, Ultimo, NSW 2007, Australia; 3Elazig Governorship, Interior Ministry, Elazig 23119, Turkey; 4Department of Management Information Systems, Management Faculty, Sakarya University, Sakarya 54050, Turkey; 5School of Computing and Information Science, Anglia Ruskin University Cambridge Campus, Cambridge CB1 1PT, UK; 6School of Science and Technology, Faculty of Science, Agriculture, Business and Law, University of New England, Armidale, NSW 2351, Australia; 7Center for Advanced Modelling and Geospatial Information Systems, Faculty of Engineering and IT, University of Technology Sydney, Sydney, NSW 2007, Australia; 8Egoscue Foundation, 12230 El Camino Real #110, San Diego, CA 92130, USA; 9Department of Digital Forensics Engineering, Technology Faculty, Firat University, Elazig 23119, Turkey; 10Ngee Ann Polytechnic, Department of Electronics and Computer Engineering, Singapore 599489, Singapore; 11Department of Biomedical Engineering, School of Science and Technology, SUSS University, Singapore 599494, Singapore; 12Department of Biomedical Informatics and Medical Engineering, Asia University, Taichung 41354, Taiwan

**Keywords:** L-tetrolet pattern, sleep stage expert system, multiple pooling decomposition, insomnia, EEG signal classification

## Abstract

Background: Sleep stage classification is a crucial process for the diagnosis of sleep or sleep-related diseases. Currently, this process is based on manual electroencephalogram (EEG) analysis, which is resource-intensive and error-prone. Various machine learning models have been recommended to standardize and automate the analysis process to address these problems. Materials and methods: The well-known cyclic alternating pattern (CAP) sleep dataset is used to train and test an L-tetrolet pattern-based sleep stage classification model in this research. By using this dataset, the following three cases are created, and they are: Insomnia, Normal, and Fused cases. For each of these cases, the machine learning model is tasked with identifying six sleep stages. The model is structured in terms of feature generation, feature selection, and classification. Feature generation is established with a new L-tetrolet (Tetris letter) function and multiple pooling decomposition for level creation. We fuse ReliefF and iterative neighborhood component analysis (INCA) feature selection using a threshold value. The hybrid and iterative feature selectors are named threshold selection-based ReliefF and INCA (TSRFINCA). The selected features are classified using a cubic support vector machine. Results: The presented L-tetrolet pattern and TSRFINCA-based sleep stage classification model yield 95.43%, 91.05%, and 92.31% accuracies for Insomnia, Normal dataset, and Fused cases, respectively. Conclusion: The recommended L-tetrolet pattern and TSRFINCA-based model push the envelope of current knowledge engineering by accurately classifying sleep stages even in the presence of sleep disorders.

## 1. Introduction

People sleep an average of eight hours a day. This shows that almost one-third of human life is spent asleep [1,2,3]. Therefore, sleep quality plays an important role in our daily life. Today, people’s sleep patterns are disrupted due to factors such as stress, intense work, and excessive use of multimedia devices [4,5,6]. Sleep disorders can negatively impact concentration, reducing task processing efficiency. Signals, such as electroencephalogram (EEG), electrocardiogram (ECG), and electrooculogram (EOG), are evaluated in people with sleep disorders. EEG signals are especially important for evaluating brain activity. EEG signals are also widely used in sleep scoring and the evaluation of sleep stages [7,8,9,10].

Two different standards are used for sleep scoring. They are the American Academy of Sleep Medicine (AASM) [11] and Rechtschaffen and Kales (R&K) [10]. The R&K standard was widely used from 1968 to 2007. Later, the sleep scoring guide was updated as the AASM standard [12].

A sleep cycle consists of the following six sleep phases: 1-W: wakefulness, 2–5-Stages (1–4): from light sleep to deep sleep, 6-REM: rapid eye movement. While the R&K standard accepts sleep stages according to this order, S3 and S4 are accepted as single stages in the AASM standard [13]. Manual identification of these stages is common during sleep disorders and sleep-related illness diagnoses [14]. This practice causes a high workload for human experts. Systems that automate sleep stage scoring are widely reported in the scientific literature [15,16,17]. These studies share the hypothesis that automated sleep stage classification can reduce the workload of human experts and ensure that errors due to environmental parameters are reduced [18,19,20]. However, automated sleep stage classification is difficult for machine learning and pattern recognition because sleep EEG datasets are heterogeneous.

We propose an L-tetrolet pattern-based sleep stage classification model that can extract transferable knowledge from heterogeneous EEG data. The popular cyclic alternating pattern (CAP) sleep EEG dataset was used to establish the sleep stage classification model. This dataset contains information from both insomniac and normal subjects, such as phase and sleep stages. Three cases were created to denote the general results of this dataset, and these cases consist of EEG signals of the insomniac subjects, normal subjects, and both insomniac and normal subjects, respectively. The proposed model could classify six sleep stages with an accuracy of 95.43%, 91.05%, and 92.31% for Insomnia, Normal dataset, and Fused cases, respectively.

Our main motivations were to propose a game-based feature extraction function and, by applying this function, present a new EEG signal classification model. To achieve that highly accurate learning model, a new L-tetrolet pattern and TSRFINCA-based sleep EEG signal classification model were created. The L-tetrolet pattern for textural feature extraction was inspired by the Tetris game. Statistical features were also extracted to enforce the presented feature generation method. A multilevel feature generation architecture was created using pooling functions to generate low-level and high-level features. The presented feature selector (TSRFINCA) incorporates three stages. In the first stage, a threshold point is determined, and feature selection is carried out by deploying this threshold point. ReliefF is applied to the selected features in the second stage, and the positive weighted feature is selected. In the last stage, iterative neighborhood component analysis (INCA) is applied to the selected features, and the most meaningful features are selected. The selected final features are utilized as the input of the cubic support vector machine (CSVM) classifier. To summarize, we proposed (i) a new game-based feature extractor, (ii) a new decomposition model by using four pooling techniques, and (iii) a hybrid high-performance feature selector. These methods have been used in a feature engineering model [21,22,23] to obtain high classification performance.

The novelties of our sleep stage classification model are given below as follows:
L-tetrolet pattern: a new, Tetris-inspired, textural feature generation function;Statistical feature generator: created by fusing multiple pooling decomposers;TSRFINCA: a three-leveled hybrid and iterative feature selector.

Contributions:
A new feature engineering model has been created by proposing new generation feature extraction, decomposition, and feature selection methods. The essential purpose of the proposed feature engineering model is to extract the most informative features from the used signals to obtain high classification performance with low time complexity. This research presents a highly accurate EEG classification model for sleep stage detection. By deploying the presented classification model, sleep stage classification results of the CAP sleep dataset are presented using three cases. Our proposal denotes general high classification performance since we applied this model to three different datasets.

The CAP Sleep Database on PhysioNet [24] is widely used in scientific work on sleep staging, and most of the published studies use the CAP database to establish the sleep phase [25,26,27,28,29,30,31,32,33]. Table 1 summarizes selected studies on sleep stage detection using different datasets.

To support our novelty claims and to substantiate the key contributions, we have structured the manuscript as follows. The next section introduces the dataset used to design and test the sleep stage classification model. Section 3 outlines the processing methods that were used to implement and test the proposed sleep stage detection model. The model was evaluated with a set of experiments. Section 4 specifies these experiments and provides the corresponding results. The subsequent discussion section relates our results to the wider sleep research area. We also list limitations and future work before concluding the paper.

## 2. Material and Method

### 2.1. Material

The CAP sleep stage dataset is a widely used benchmark dataset. The dataset consists of EEG recordings during the Non-REM (NREM) sleep phase. These data were obtained from 108 polysomnographic patients registered at the Sleep Disorders Center of the Ospedale Maggiore of Parma in Italy [24]. The data were recorded as .edf files [47]. Sleep data comprises at least three EEG channels, two EOGs, submentalis muscle EMG, bilateral anterior tibial EMG, respiratory signals, and ECG. In total, 16 of the subjects were healthy, and 92 were pathological. Table 2 shows the neurological status and number of subjects [44]. The age range of the subjects is 14–82, and the average age is 45. In total, 61% of the subjects were men (66 people), and 38% were women (42 people).

The CAP Sleep Database has been downloaded from Physionet [48]. Expert neurologists labeled these sleep data according to Rechtschaffen & Kales (R&K) rules using the sleep stage (W = waking, S1–S4 = sleep stages, R = REM, MT = body movements), time, duration, signal type data in the tag files. Each label classifies a unique (non-overlapping) 30-s data window. Data start time, hypnogram start time, and frequency information are needed for labeling. This information was obtained from files with EDF extensions. Using the Matlab 2019b program, the .edf files were read, and all recorded channels were listed. Of these channels, only the F4-C4 channels were used. Due to the absence of F4-C4 (these channels are commonly used EEG channels [43,49,50]. Therefore, we used these channels.) channels, some of the normal recordings were ignored.

### 2.2. Method

This research presents a new, handcrafted feature-based EEG signal classification model. Feature creation, feature selection, and classification are the main phases of the presented model. The feature creation step incorporates both textural and statistical methods. Maximum pooling was used to create decomposed signals. By using the created decomposed signals, features have been extracted at both low and high levels. Specifically, we used absolute pooling, average pooling, and maximum absolute pooling. In the feature selection phase, a three leveled selector (TSRFINCA) was employed. In the classification phase, CSVM was deployed as a classifier. The general steps of this model are given below. 

Step 0: Load EEG signals.

Step 1: Apply absolute average pooling, average pooling, and absolute maximum pooling to obtain M1 (absolute average pooling), M2 (average pooling), and M3 (absolute maximum pooling) signals. Herein, we used non-overlapping blocks with a length of two to create decompressed signals. In the M1 function, the absolute value of the used non-overlapping block was used as the decompressed signal. In M2 and M3, we used the maximum of the absolute block values and average values of the used non-overlapping blocks for decomposition. Equations (3)–(8) provide a mathematical definition of these functions.

Step 2: Extract 512 textural features from each signal (raw EEG signal and the generated M1, M2, and M3 signals). In this step, 4 × 512 = 2048 features have been generated.

Step 3: Generate 36 statistical features from each signal and textural features by using 18 statistical moments. The used 18 statistical moments have been applied to the raw signal and the generated textural features in Step 2. 

Two main feature extraction methodologies, namely, e-textural and statistical feature extraction, were used for handcrafted feature extraction. By deploying our proposed L-tetrolet pattern, textural features were generated. Statistical features were extracted using statistical moments to enforce our feature generation phase.

Step 4: Apply maximum pooling to the EEG signal and update signal. This step defines the decomposition level.

Step 5: Repeat Steps 1–4 five times. Herein, a multilevel feature generator is created. By using handcrafted feature extractors, only low-level features have been generated. To create high-level features, a multilevel feature extraction model was created. Equation (1) provides a mathematical definition of the maximum pooling operator.
(1)$$D=MaxP\left(EEG\right)$$
(2)$$D\left(j\right)=\max\left(\left[EEG\left(i\right)\;EEG\left(i+1\right)\right]\right),\;j\in \left\{1,2,\ldots ,|\frac{L}{2}|\right\},i\in \left\{1,3,\ldots ,L-1\right\}\;$$

Herein, *MaxP*(.) defines the maximum pooling function, *D* is decomposed signal, *L* is the length of the used EEG signal (*EEG*), and *max*(.) is maximum value finding function.

Step 6: Fuse the generated features. 

Step 7: Summarize each feature individually.

Step 8: Determine the threshold point to eliminate redundant features.

Step 9: Apply ReliefF [51] to features and generate a weight for each feature.

Step 10: Choose positive ReliefF [51] weighted features. 

Step 11: Apply INCA [52] to the positive weighted feature by selecting ReliefF in Step 10.

Step 12: Forward the selected features to the classifier.

The twelve steps detailed above define the proposed decision support model. Steps 1–6 represent the L-tetrolet feature generation. Steps 8–11 denote TSRFINCA feature selection, and Step 12 demonstrates the classification phase. Figure 1 shows the proposed L-tetrolet pattern-based sleep stage classification model flow diagram. The next sections introduce the individual model phases in detail.

#### 2.2.1. L-Tetrolet Pattern and Statistical Features Based Multileveled Feature Generation Method

Feature generation/extraction is the first phase of the proposed decision support method. Statistical and textural features were generated in this phase. Linear and nonlinear statistical moments were used to generate statistical features, and 18 statistical features were generated by using these moments. In the textural feature generation phase, we present a new microstructure that was inspired by the Tetris game. The letter ‘L’ (L-tetrolet) of the Tetris game was employed for pattern identification [53,54]. Therefore, the presented textural feature generation function is called an L-tetrolet pattern. The L-tetrolet pattern generates 512 features from a one-dimensional signal. Statistical features were also extracted from the generated textural features by deploying the 18 moments. 

The primary objective of the presented feature generation model is to create low-level and high-level features. Therefore, a multileveled/multilayered method was employed to generate these features. A pooling-based decomposer was utilized as a decomposition method. By deploying four pooling functions – absolute average pooling, absolute maximum pooling, average pooling, and maximum pooling –, a five leveled feature generation method was created. The steps of the presented feature generation method are given below.

Step 1: Employ average, absolute maximum, and absolute average pooling to decompose the raw EEG signal into M1, M2, and M3. Here, 2 size non-overlapping blocks were used.
(3)$$M1=avp\left(EEG\right)$$
(4)$$M2=avp^{ab}\left(EEG\right)$$
(5)$$M3=max^{ab}\left(EEG\right)$$
(6)$$avp\left(EEG\right)=M1\left(j\right)=\frac{EEG\left(i\right)+EEG\left(i+1\right)}{2},\;i=\left\{1,3,\ldots ,Ln-1\right\},j=\left\{1,2,\ldots ,|\frac{Ln}{2}|\right\}$$
(7)$$avp^{ab}\left(EEG\right)=M2\left(j\right)=\frac{\left|EEG\left(i\right)\left|+\right|EEG\left(i+1\right)\right|}{2},i=\left\{1,3,\ldots ,Ln-1\right\}\;$$
(8)$$max^{ab}\left(EEG\right)=M3\left(j\right)=\left\{\begin{array}{c} \left|EEG\left(i\right)\right|,\;\left|EEG\left(i\right)\right|\geq \left|EEG\left(i+1\right)\right|\\ \left|EEG\left(i+1\right)\right|,\;\left|EEG\left(i\right)\right|<\left|EEG\left(i+1\right)\right| \end{array}\right.\;,\;\;i=\left\{1,3,\ldots ,Ln-1\right\}$$
where $avp\left(.\right),avp^{ab}\left(.\right),\;maxp^{ab}\left(.\right)$ define average pooling, absolute average pooling, and absolute maximum pooling. EEG denotes the one-dimensional measurement signal, Ln represents the signal length. $\left|.\right|$ is absolute function. 

Step 2: Generate features from the generated M1, M2, M3, and the raw one-dimensional signal (EEG). In this step, both statistical moments and the presented L-tetrolet pattern were used.
(9)$$f^{st}=st\left(EEG\right)$$
(10)$$f^{T}=L-tetrolet\left(EEG\right)$$
(11)$$f^{stT}=st\left(L-tetrolet\left(EEG\right)\right)$$

In these Equations (see Equations (7) and (8)) statistical feature generation function (*st*(.)) and L-tetrolet pattern (*L* − *tetrolet*(.)) are defined. *f^st^* represents 18 statistical features, *f^T^* is 512 textural features and *f^stT^* is the statistical features of the generated textural features. Table 3 lists the statistical moments that were used for feature extraction [55].

Here, the used 12th, 16th, 17th, and 18th moments extract nonlinear statistical features.

The presented L-tetrolet pattern was used to extract textural features. The steps of this function are detailed as follows:

Step 2.1: Divide the used one-dimensional signals into overlapping blocks/windows (blk) with a size of 16.
(12)$$blk=EEG\left(i+t-1\right),\;i=\left\{1,2,\ldots ,Ln-15\right\},\;t=\left\{1,2,\ldots ,16\right\}$$

Step 2.2: Create a matrix (mtr) with a size of 4 × 4 using the constructed block.
(13)$$mtr\left(k,l\right)=blk\left(t\right),\;k=\left\{1,2,3,4\right\},\;l=\left\{1,2,3,4\right\}$$

Figure 2 depicts the resulting 4 × 4 matrix.

Step 2.3: Use two L-tetrolet based patterns by employing the 4 × 4 sized matrix. Figure 3 shows the L-tetrolet patterns that were used for feature generation.

Step 2.4: Extract bits using P1, P2, and binary feature generation function $S\left(.,.\right)$. These patterns (P1 and P2) are separately applied to the generated matrix. For P1 and P2, the used a, b, c, and d values are given in Equation (14) according to Figure 2 and Figure 3.
(14)$$\left|\begin{array}{cc} a^{1}\left(1\right) & a^{2}\left(1\right)\\ a^{1}\left(2\right) & a^{2}\left(2\right)\\ a^{1}\left(3\right) & a^{2}\left(3\right)\\ a^{1}\left(4\right) & a^{2}\left(4\right)\\ b^{1}\left(1\right) & b^{2}\left(1\right)\\ b^{1}\left(2\right) & b^{2}\left(2\right)\\ b^{1}\left(3\right) & b^{2}\left(3\right)\\ b^{1}\left(4\right) & b^{2}\left(4\right)\\ c^{1}\left(1\right) & c^{2}\left(1\right)\\ c^{1}\left(2\right) & c^{2}\left(2\right)\\ c^{1}\left(3\right) & c^{2}\left(3\right)\\ c^{1}\left(4\right) & c^{2}\left(4\right)\\ d^{1}\left(1\right) & d^{2}\left(1\right)\\ d^{1}\left(2\right) & d^{2}\left(2\right)\\ d^{1}\left(3\right) & d^{2}\left(3\right)\\ d^{1}\left(4\right) & d^{2}\left(4\right) \end{array}\right|=\left|\begin{array}{cc} V_{1} & V_{1}\\ V_{5} & V_{5}\\ V_{9} & V_{6}\\ V_{10} & V_{7}\\ V_{4} & V_{8}\\ V_{3} & V_{4}\\ V_{2} & V_{3}\\ V_{6} & V_{2}\\ V_{16} & V_{9}\\ V_{12} & V_{13}\\ V_{8} & V_{14}\\ V_{7} & V_{15}\\ V_{13} & V_{16}\\ V_{14} & V_{12}\\ V_{15} & V_{11}\\ V_{11} & V_{10} \end{array}\right|$$

Herein, $a^{1},b^{1},c^{1},d^{1}$ are belonging to P1 pattern and $a^{2},b^{2},c^{2},d^{2}$ are belonging to P2 pattern. By using these values, feature extraction process has been conducted. The bit generation phase has been given below.
(15)$$bit^{t}\left(k\right)=S\left(a^{t}\left(k\right),c^{t}\left(k\right)\right),\;k=\left\{1,2,3,4\right\},\;t=\left\{1,2\right\}$$
(16)$$bit^{t}\left(k+4\right)=S\left(b^{t}\left(k\right),d^{t}\left(k\right)\right)$$
(17)$$S\left(par1,par2\right)=\left\{\begin{array}{c} 0,\;par1-par2<0\\ 1,\;par1-par2\geq 0 \end{array}\right.$$
where par1 and par2 are the first and second parameters of the binary feature generation (signum) function. Equations (15)–(17) were deployed to both P1 and P2, and eight bits were extracted from each pattern. The extracted bits are named $bit^{1}$ and $bit^{2}$ (they are shown using $bit^{t}$ in Equations (15) and (16)). The length of each bit array is equal to eight. By deploying these bits, two novel signals were created for feature generation, and these signals were named the first map (map1) and the second map signal (map2), respectively. Binary to decimal conversion was used to create these signals, as shown in Equations (18) and (19). 

Step 2.5: Create map signals employing the generated bits.
(18)$$map1\left(i\right)=\overset{8}{\underset{k=1}{\sum }}bit^{1}\left(k\right)*2^{k-1}$$
(19)$$map2\left(i\right)=\overset{8}{\underset{k=1}{\sum }}bit^{2}\left(k\right)*2^{k-1}$$

Step 2.6: Extract histograms of the map1 and map2 signals. Each histogram has $2^{8}=256$ values.
(20)$$hist^{1}=\delta \left(map1\right)$$
(21)$$hist^{2}=\delta \left(map2\right)$$
where $hist^{1}$ and $hist^{2}$ are histograms of the first and second map signals, respectively. $\delta \left(.\right)$ function is defined to extract histogram.

Step 2.7: Create a feature vector (feat) with a length of 512 by using $hist^{1}$ and $hist^{2}$.
(22)$$feat\left(h\right)=hist^{1}\left(h\right),\;h=\left\{1,2,\ldots ,256\right\}$$
(23)$$feat\left(h+256\right)=hist^{2}\left(h\right)$$

Equations (19) and (20) define the feature concatenation process.

The given steps above (see Steps 2.1–2.7) are defined our proposed L-tetrolet pattern.

Step 3: Merge the generated textural, statistical, and statistical textural features of each signal. For a one-dimensional signal, 512 + 18 + 18 = 548 features were generated. In a level, the defined feature generation functions were applied to four signals (M1, M2, M3, and raw signal). Therefore, these functions generate 548 × 4 = 2192 features at each level.

Step 4: Decompose the one-dimensional signal (EEG) by deploying the maximum pooling decomposer. This step defines signal updating. 

Step 5: Repeat Steps 1–4 five times utilizing decomposed signal input. This constitutes the multilevel feature extractor. 

Step 6: Merge generated features in each level and obtain 2192 × 5 = 10,960 features from a one-dimensional signal.

#### 2.2.2. Threshold Selection Based Relieff and Iterative Neighborhood Component Analysis

A three-layered feature selection model was used in this phase, and these layers were threshold-based feature selection, positive ReliefF weighted features selection, and INCA selection processes. The primary objectives of this feature selector were the following:
Present an effective feature selector;Use advantages of the three feature selection methods together;Select the most appropriate features automatically.

Figure 4 shows a block diagram of the proposed TSRFINCA selector.

The following steps introduce the TSRFINCA functionality:

Step 1: Normalize the generated features (X) individually.
(24)$$X^{norm}\left(:,i\right)=\frac{X\left(:,i\right)-min\left(X^{norm}\left(:,i\right)\right)}{max\left(X^{norm}\left(:,i\right)\right)-min\left(X^{norm}\left(:,i\right)\right)},\;i=\left\{1,2,\ldots ,10960\right\}$$
where $X^{norm}$ represents normalized features by deploying min-max normalization.

Step 2: Deploy threshold-based feature selection. In this study, we used zero as threshold (β). The mathematical descriptions of this method are given below.
(25)$$tpl\left(j\right)=\overset{D}{\underset{d=1}{\sum }}X^{norm}\left(d,j\right),j=\left\{1,2,\ldots ,10960\right\}$$
(26)$$X^{1}\left(:,cnt\right)=X^{norm}\left(:,j\right),\;cnt=cnt+1,\;if\;tpl\left(j\right)>\beta$$
where tpl means summarization of the features, $X^{1}$ is the selected features in the first layer, and cnt is a counter.

Step 3: Employ ReliefF to $X^{1}$ and generate ReliefF weights ($w^{RF}$).

Step 4: Eliminate negative weighted features to obtain second layer features ($X^{2}$).
(27)$$X^{2}\left(:,cnt\right)=X^{1}\left(:,j\right),\;cnt=cnt+1,\;if\;w^{RF}\left(j\right)>0$$

Step 5: Apply INCA to $X^{2}$ and obtain the final features ($X^{3}$). 

INCA is an iterative selector, and it can select features of various sizes and hence it is applicable to a wide range of problem solutions. In this work, we progress now to the classification algorithm that was used for sleep stage detection.

#### 2.2.3. Classification

Classification is the last phase of the presented sleep stage classification model. Here, we used a CSVM classification algorithm. The hyper-parameters of this classifier are given below as follows:
Training and testing method: 10-fold cross-validation;Kernel: Third-degree polynomial order (Cubic);Box constraint level (C value): One;Multiclass method: One-vs-one.

## 3. Results

### 3.1. Experimental Setup

The CAP dataset was downloaded from Physionet to train and test the presented L-tetrolet pattern and TSRFINCA-based sleep stage classification model. This research focused on the sleep stages of insomniacs and normal subjects. The sleep stage datasets are generally heterogeneous. Therefore, high classification rates do not reflect the model performance. A balanced EEG dataset has been created to overcome this problem by randomly selecting EEG signals from each subject. By creating these three datasets, the following three cases were defined, and these are explained below:

Case 1: This dataset was collected from the insomnia subjects. It includes the following six classes: wake, stage 1, stage 2, stage 3, stage 4, and REM. This dataset contains 1356 EEG signals (each class has 226 EEG signals). F4-C4 channels have been used in this case.

Case 2: This case uses EEG signals from normal subjects. A homogenous dataset was created in this case. There are 1698 EEG signals in this dataset (each class has 283 EEG signals). F4-C4 channels have been used in this case.

Case 3: In this case, a merged dataset is used. This dataset was created by merging datasets of Cases 1–2. Therefore, it contains 3054 EEG signals (each class has 283 + 226 = 509 EEG signals). F4-C4 channels have been used in this case.

These three balanced datasets were used to define three distinct sleep stage identification tasks. The MATLAB (2020a) programming environment was used to calculate test results and implement the proposed decision support model. The used functions were named main, L-tetrolet pattern, statistical feature generator, TSRFINCA, and classification. In the main function, the EEG signals were read, and other functions were called in the main function to classify sleep stages. The proposed model was implemented on a basic desktop computer, and parallel programming or hardware acceleration was not used.

### 3.2. Results

The model quality was obtained by assessing the classification results according to the rules of 10-fold cross-validation. Here, six classification results were presented. Accuracy, F1-score, average precision, and geometric mean results were calculated. Table 4 lists the calculated results for each case.

As can be seen from Table 4, the recommended method yielded 95.43%, 91.05%, and 92.31% classification accuracies for Case 1, Case 2, and Case 3, respectively. A 10-fold cross-validation was used to calculate these results. Table 5 details the fold-by-fold results.

### 3.3. Computational Complexity Analysis

Computational complexity is a crucial property that determines the practicality of the proposed model. A lower computational complexity is more resource-efficient, which translates into less energy usage and lower cost. The presented model consists of three algorithms. Therefore, the time complexities of these algorithms should be calculated [56,57]. Table 6 introduces these calculations in detail.

In this table (see Table 6), the used coefficients are given as follows. n is the length of the signal, d defines the number of observations, k represents the time complexity coefficient of the used feature selection and classification models, and I defines the number of iterations for iterative feature selection.

Feature generation: We have used a multileveled feature generation in this study. In each level, a maximum pooling decomposer was used to halve the signal length. The used feature generation functions (L-tetrolet pattern and statistical feature generator) have low computational complexity ($O\left(n\right)$). Therefore, the time complexity of this phase is calculated as $O\left(ndlognd\right)$. Here, n is the size of the EEG signal, and d represents the number of EEG signals.

Feature selection: The TSRFINCA algorithm has three layers. The threshold-based feature selection model is a simple and basic model. Therefore, the time complexity of it is calculated as $O\left(kd\right)$. Here, *k* is defined as the number of features. In this phase, INCA is the most complex feature selector and $O\left(Ik^{3}d\right)$ is found as the computational complexity. Here, I is the number of iterations because it is an iterative feature selector, and in each iteration, the loss value is calculated using the SVM classifier.

Classification: A CSVM classifier, with a time complexity of $O\left(k^{3}d\right)$, was employed for classification.

As can be seen from the time complexity analysis (see Table 6), the proposed model has a low time burden. The deep learning models have an exponential time burden, but this model has a linear time burden. Therefore, there is no need to use extra hardware to implement our proposal. Furthermore, this model can extract features at low and high levels.

## 4. Discussion

As stated in Sect. 3, the presented model has the following three fundamental phases: feature generation, TSRFINCA-based feature selection, and classification. The presented model uses four pooling methods to overcome the routing problem of pooling. For instance, the maximum pooling only routes peak values. We proposed a multiple pooling-based decomposition model to overcome the routing problem. Both textural and statistical feature generators have been utilized for feature extraction to create handcrafted features. By using these feature extractors and the proposed multiple pooling function, a multileveled feature extraction method has been presented to generate features at both low and high levels. The presented three-layered feature selection function—TSRFINCA—selected the top informative features from the used datasets. In this research, we have used three datasets. The presented TSRFINCA selects a variable-sized feature vector for each dataset. The sizes of the optimal feature sets were found to be 644, 711, and 188 for Case 1, Case 2, and Case 3, respectively. Figure 5 documents the feature selection process.

In this figure (see Figure 5), a number of features and loss values are demonstrated. The proposed feature selector is an iterative feature selector and it calculates loss values of the 901 (initial value and end value of the loop are 100 and 1000, respectively; thus, 901 = 1000 − 100 + 1 feature vectors have been evaluated for each dataset) feature vectors. The optimal feature vectors have been selected using minimum loss values. These optimal features were forwarded to the CSVM classifier. This classifier was utilized as both a loss value generator (calculating misclassification rates of the chosen 901 feature vectors) and a classifier. Figure 6 shows a confusion matrix for each case.

The confusion matrices in Figure 6 denote the case-specific results.

To select the optimal classifiers, features of Case 3 were tested on the shallow variable classifiers. These were decision tree (DT) [58], linear discriminant (LD) [59], Naïve Bayes (NB) [60], linear SVM (LSVM) [61], CSVM [62], quadratic SVM (QSVM) [62], k nearest neighbors (kNN) [63] and bagged tree (BT) [64]. Figure 7 introduces the accuracies achieved with the individual classifiers.

Figure 6 demonstrates that the best classifier is CSVM. Therefore, CSVM is selected as both an error generator and a classifier.

We have compared our model with other sleep stage classification methods. Table 7 lists the comparison results.

Table 7 shows the success of the presented L-tetrolet pattern and TSRFINCA-based model. Moreover, prior presented models generally used a single dataset, but we tested our model on three balanced/homogenous datasets. Our proposal attained over 90% classification accuracy for all cases. These findings clearly demonstrate our success. The advantages of this model are the following:A new game-inspired feature generation model is presented, and the effectiveness of this approach is established through EEG-based sleep stage classification;To overcome the routing problem of the pooling method, a multiple pooling decomposer-based feature generation strategy was used;A three-layered feature selector is presented;By applying these methods and CSVM, a highly accurate sleep stage classification model is presented;The recommended model outperformed;The proposed model can be applied to a computer with basic system configurations.

The drawbacks of this research are the following:The presented TSRFINCA is a hybrid and iterative feature selector, but the computational complexity is high. Moreover, we have used a shallow classifier. In this work, deep classifiers can be used to increase the classification ability, or a metaheuristic optimization model can be used to tune the hyperparameters of the used classifier;The datasets used are small. Therefore, when we used one dataset for training and the other datasets for testing, we achieved a classification accuracy of about 50%. Since these EEG signals have sick subjects (each case defines a disorder).

Diagnoses of sleep or sleep-related diseases are time-consuming because the diagnostic pathway relies on manual signal analysis. A new EEG-based sleep stage detection/monitoring system can be developed soon to help medical professionals with diagnosis. Figure 8 denotes the intended intelligent monitoring system.

This research presents a new game-based feature generation function. The L-tetrolet pattern is inspired by the Tetris game. Other game-based feature generation or decomposition models can be presented in future studies, and the recommended model can be applied to other one-dimensional signals to solve classification problems. In the future, we plan to develop a game-based deep learning model for one-dimensional signal classification, which might replace or augment recurrent neural networks.

## 5. Conclusions

In this research, we propose a new feature engineering model. The essential goal of that model is to extract the most significant features from EEG signals. The model is based on a new game-based feature extraction function, named L-tetrolet, which extracts textural feature information. To generate high-level features, a multileveled feature extraction structure is presented using a combination of four pooling techniques. This approach fuses hybrid approximation and the advantages of pooling techniques. In the feature selection phase, a three-layered hybrid feature selector has been used, and the selected features have been classified using a shallow classifier. Using PhysioNet, three different sleep EEG datasets were created, each containing six groups. Our proposed L-tetrolet-based model attained >90% overall classification accuracy on these datasets. Moreover, our proposal reached 95.43% classification accuracy in Case 1. These results were compared to other recent models, showing that our model outperforms all the previous methods used for sleep stage detection based on signals from the CAP database. These findings demonstrated that our model achieved satisfactory classification performance and time complexity for solving sleep stage classification problems using EEG signals.

In the future, we plan to accomplish the following:-Propose new game-based feature extraction functions;-Purpose self-organized feature engineering models;-Propose a new generation of pooling/decomposition methods by using quantum computing and superposition;-Develop a new sleep stage classification application, which will be used in medical centers.

## Figures and Tables

**Figure 1 diagnostics-12-02510-f001:**
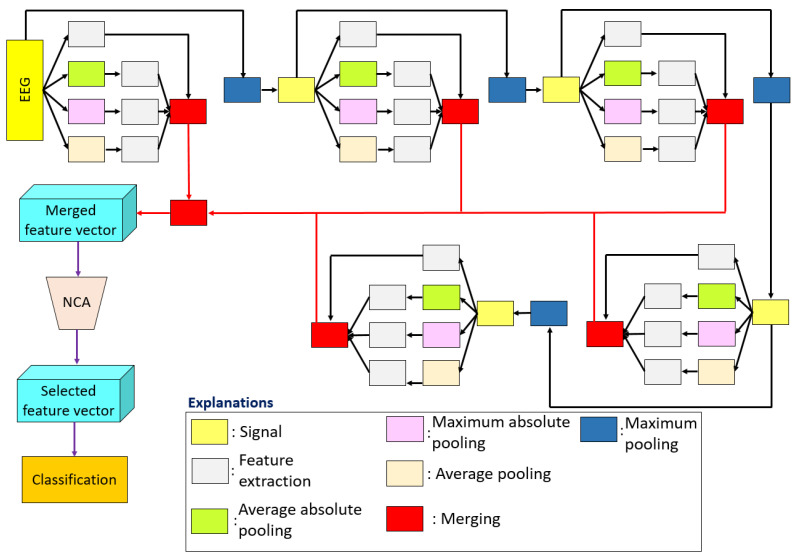
Snapshot of the proposed L-tetrolet and TSRFINCA based sleep stage classification model.

**Figure 2 diagnostics-12-02510-f002:**
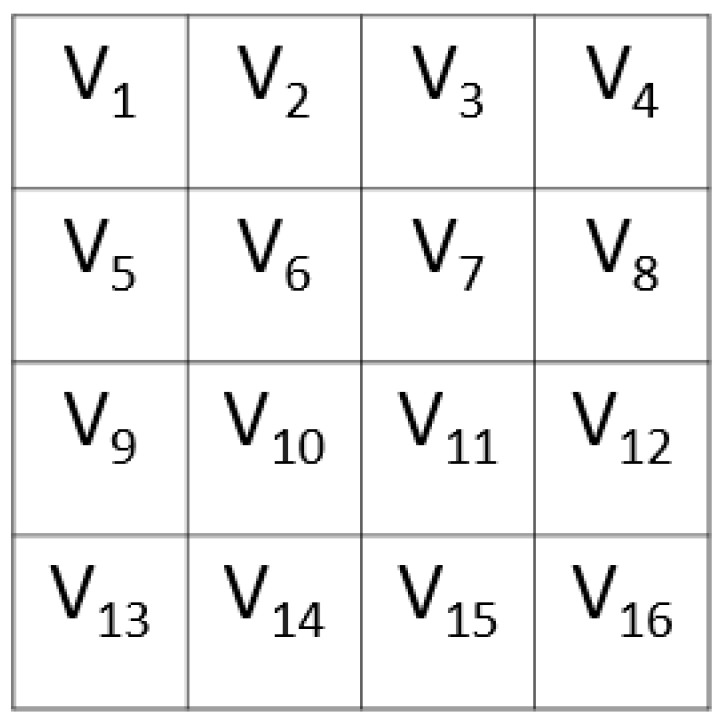
The 4 × 4 matrix that was created for applying the proposed L-tetrolet pattern.

**Figure 3 diagnostics-12-02510-f003:**
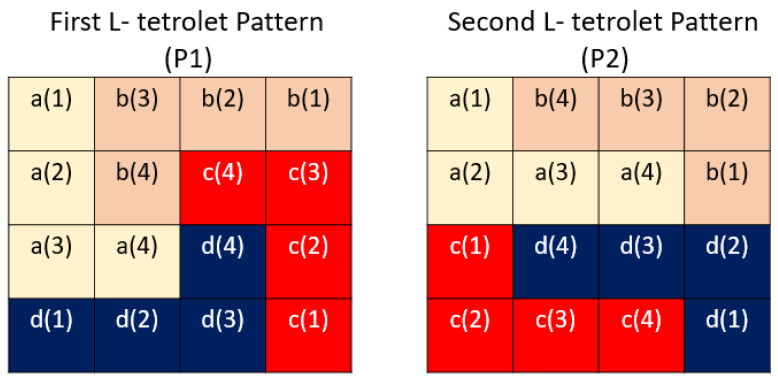
The used L-tetrolet patterns. Each L-tetrolet is named using a letter (e.g., a, b, c, d), and these letters are shown using different colors. These patterns are called P1 and P2.

**Figure 4 diagnostics-12-02510-f004:**
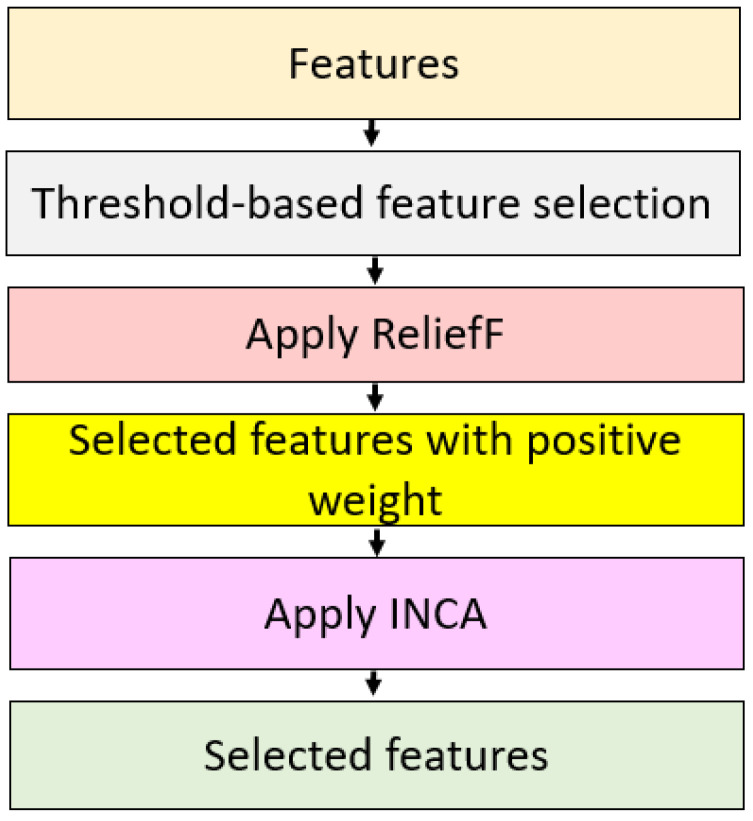
Block diagram of the TSRFINCA model.

**Figure 5 diagnostics-12-02510-f005:**
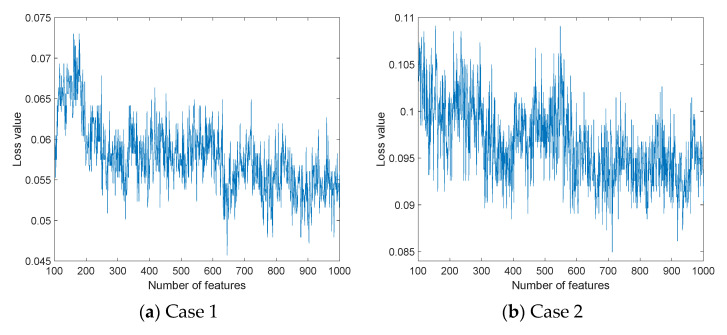
Feature selection processes of the cases.

**Figure 6 diagnostics-12-02510-f006:**
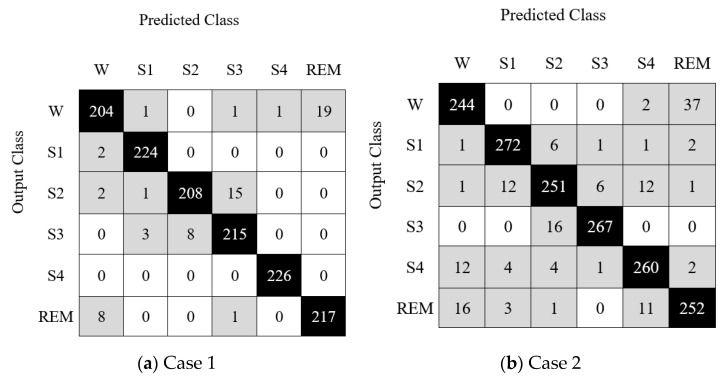
Confusion matrix for each case.

**Figure 7 diagnostics-12-02510-f007:**
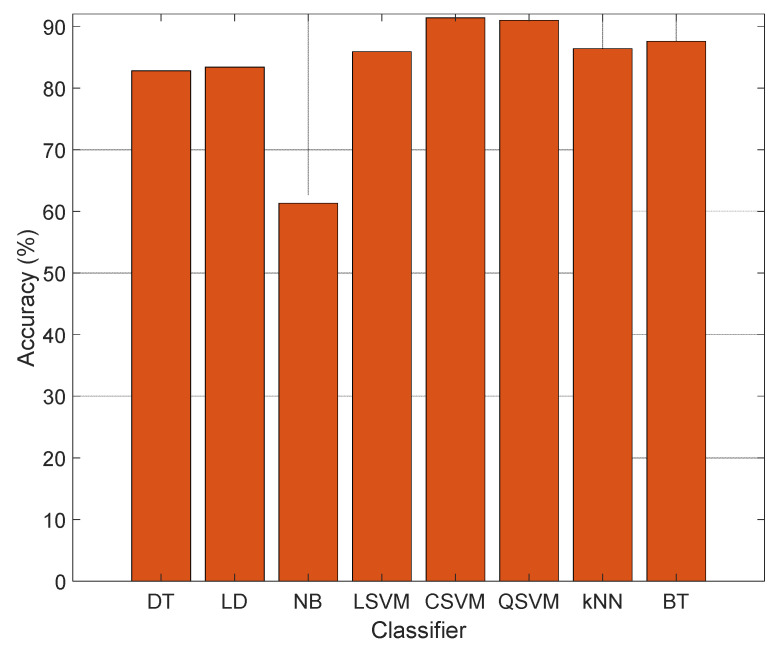
Classification accuracies of the classifier. Here, the presented L-tetrolet and maximum pooling-based feature generation method is applied to Case 3. The first and second layers of the TSRFINCA are applied to these features to eliminate the redundant feature, and NCA selected 1000 features for tests.

**Figure 8 diagnostics-12-02510-f008:**
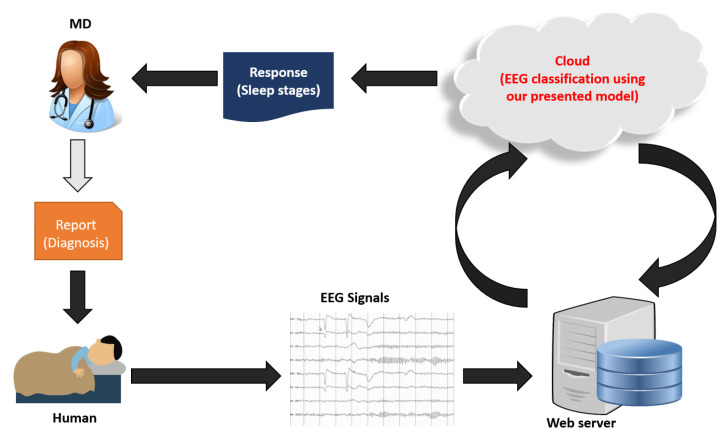
The intended automated sleep stage classification and monitoring model.

**Table 1 diagnostics-12-02510-t001:** Literature review on sleep stage detection.

Studies	Method	Classifier	Dataset	Channels	The Results (%)
Abbasi et al. [34]	Convolutional Neural Network	Ensemble	Collected data	Multiple channels	Sensitivity: 78.44 Specificity: 96.49 Accuracy: 94.27
Li et al. [35]	Multi-Layer Convolutional Neural Networks	Auxiliary	SHHS dataset [36]	C3-A2, C4-A1, EOG	Accuracy: 85.12
Zaidi and Farooq [37]	Fourier Synchrosqueezed Transform Features	Support vector machine	DREAMS dataset	Cz-A1	Accuracy: 82.60
Sors et al. [38]	Deep Convolutional Neural Network	Convolutional Neural Network	The Sleep Heart Health Study dataset [39]	C4-A1, C3-A2	Accuracy: 87.00
Goshtasbi et al. [40]	Convolutional Neural Network	Softmax	SHHS dataset [36]	C4-A1, C3-A2	Accuracy: 81.30 Kappa: 74.00
Shahbakhti et al. [41]	Nonlinear Analysis	Linear discriminant analysis	DREAMS dataset [42]	Fp1, O1, and CZ or C3	Accuracy: 92.50 Sensitivity: 89.90 Specificity: 94.50
Zhao et al. [43]	SleepContextNet	Softmax	1. SHHS dataset [36] 2. CAP dataset [24,44]	C4-A1 and C3-A2	1. Accuracy: 86.40 Kappa: 81.00 2. Accuracy: 78.80 Kappa: 71.00
Eldele et al. [45]	Multi-Resolution Convolutional Neural Network, Adaptive Feature Recalibration	Softmax	SHHS dataset [36]	C4-A1	Accuracy: 84.20 Kappa: 78.00
Yang et al. [46]	One-Dimensional Convolutional Neural Network, Hidden Markov model	One-Dimensional Convolutional Neural Network, Hidden Markov model	DRM-SUB dataset [42]	Pz-Oz	Accuracy: 83.23 Kappa: 76.00

**Table 2 diagnostics-12-02510-t002:** Neurological status and number of subjects.

The Neurological Status	F	M	Age: Min–Max (Average)	Number of Patients
No pathology (controls/normal)	9	7	23–42 (32.18)	16
Nocturnal frontal lobe epilepsy (NFLE)	19	21	14–67 (30.27)	40
REM behavior disorder (RBD)	3	19	58–82 (70.72)	22
Periodic leg movements (PLM)	3	7	40–62 (55.10)	10
Insomnia	5	4	47–82 (60.88)	9
Narcolepsy	3	2	18–44 (31.60)	5
Sleep-disordered breathing (SDB)	-	4	65–78(71.25)	4
Bruxism	-	2	23–34 (28.50)	2
**Total number of pathologies**	**33**	**59**	**14–82 (49.19)**	**92**

F: female, M: male.

**Table 3 diagnostics-12-02510-t003:** The statistical moments used for the generation of statistical features.

Num	Equation	Num	Equation
1	$\frac{1}{Ln}\overset{Ln}{\underset{j=1}{\sum }}EEG\left(j\right)$	10	$max\left(EEG\right)-median\left(EEG\right)$
2	$\sqrt{\frac{\sum_{i=1}^{Ln}(EEG\left(i\right)-\frac{1}{Len}\sum_{j=1}^{Len}EEG\left(j\right))}{Ln-1}}$	11	$\frac{1}{Ln}\overset{Ln}{\underset{j=1}{\sum }}\left|EEG\left(j\right)\right|$
3	$max\left(EEG\right)$	12	$-\overset{Ln}{\underset{j=1}{\sum }}log{\left(prb\left(EEG\left(j\right)\right)\right)}^{2}$
4	$min\left(EEG\right)$	13	$max\left(\left|EEG\right|\right)-min\left(\left|EEG\right|\right)$
5	$median\left(EEG\right)$	14	$min\left(\left|EEG\right|\right)$
6	$\frac{1}{Ln}{\left(\overset{Ln}{\underset{i=1}{\sum }}(EEG\left(i\right)-\frac{1}{Ln}\overset{Ln}{\underset{j=1}{\sum }}EEG\left(j\right))\right)}^{2}$	15	$\sqrt{\frac{\sum_{i=1}^{Ln}(\left|EEG\left(i\right)\right|-\frac{1}{Ln}\sum_{j=1}^{Ln}\left|EEG\left(j\right)\right|)}{Ln-1}}$
7	$\frac{1}{Ln}\overset{Ln}{\underset{j=1}{\sum }}EEG{\left(j\right)}^{2}$	16	$-\overset{Ln}{\underset{j=1}{\sum }}prb\left(EEG\left(j\right)\right)*log\left(prb\left(S\left(j\right)\right)\right)$
8	$\frac{1}{Ln}\overset{Ln}{\underset{i=1}{\sum }}|EEG\left(i\right)-\frac{1}{Ln}\overset{Ln}{\underset{j=1}{\sum }}EEG\left(j\right)|$	17	$\overset{Ln}{\underset{j=1}{\sum }}EEG{\left(j\right)}^{2}$
9	$max\left(EEG\right)-min\left(EEG\right)$	18	$-\overset{Ln}{\underset{j=1}{\sum }}prb{\left(EEG\left(j\right)\right)}^{2}*log{\left(prb\left(EEG\left(j\right)\right)\right)}^{2}$

where $prb\left(.\right)$ defines probability.

**Table 4 diagnostics-12-02510-t004:** The calculated performance results of the presented L-tetrolet pattern and TSRFINCA model.

Case	Accuracy	F1-Score	Average Precision	Geometric Mean	Sensitivity	Specificity
Case 1	95.43%	95.42%	95.46%	95.36%	90.27	98.94
Case 2	91.05%	90.01%	90.08%	89.95%	86.22	97.17
Case 3	92.31%	92.29%	92.29%	92.23%	87.03	97.96

**Table 5 diagnostics-12-02510-t005:** Fold-by-fold accuracies in % for the three cases.

Fold	Case 1	Case 2	Case 3
Fold-1	86.03	80.59	84.26
Fold-2	97.06	94.12	92.46
Fold-3	100.0	98.24	95.74
Fold-4	88.97	93.53	93.77
Fold-5	97.06	85.29	87.87
Fold-6	94.85	95.88	95.08
Fold-7	97.06	86.47	91.80
Fold-8	100.0	95.88	98.36
Fold-9	98.53	90.00	91.48
Fold-10	94.70	90.48	92.23
Overall	95.43	91.05	92.31

**Table 6 diagnostics-12-02510-t006:** The time complexity calculation of the presented model.

Phase	Steps	Computational Complexity
Feature generation	Pooling-based decomposition	$O\left(ndlognd\right)$
	Statistical feature generation	$O\left(ndlognd\right)$
	Textural feature generation (L-tetrolet pattern)	$O\left(ndlognd\right)$
	Statistical features extraction of the textural features	$O\left(ndlognd\right)$
TSRFINCA	Threshold feature selection	$O\left(kd\right)$
	ReliefF-based selection	$O\left(kd\right)$
	INCA	$O\left(Ik^{3}d\right)$
Classification	SVM	$O\left(k^{3}d\right)$
Total	$O\left(4ndlognd+2kd+Ik^{3}d+k^{3}d\right)≅O\left(ndlognd+Ik^{3}d\right)$	

**Table 7 diagnostics-12-02510-t007:** The comparison results.

Study	Dataset	Accuracy Result (%)
Bajaj and Pachori [65]	Sleep-EDF dataset [24,66]	88.47 (Pz-Oz)
Hassan et al. [67]	Sleep-EDF database [24,66]	90.69 (Pz-Oz)
Jiang et al. [68]	1. Sleep-EDF database [24,66] 2. Sleep-EDF Expanded database [24]	89.40 (Fpz-Cz) 88.30 (Pz-Oz)
Kanwal et al. [69]	Sleep-EDF database [24,66]	93.00 (Pz-Oz, PFz-Cz, EOG)
Basha et al. [70]	Sleep-EDF database [24,66]	90.20 (PFz-Cz)
Jadhav et al. [71]	Sleep-EDF Expanded database [24]	85.07 (PFz-Cz) 82.92 (Pz-Oz)
Michielli et al. [72]	Sleep-EDF database [24,66]	90.80 (Pz-Oz)
Huang et al. [73]	Sleep-EDF Expanded database [24]	84.60 (Fpz-Cz) 82.30 (Pz-Oz)
Kim et al. [74]	CAP Sleep Database on PhysioNet [24]	73.60 (unspecified)
Shanin et al. [75]	Collected data	92.00 (C3-C4)
Karimzadeh et al. [76]	Sleep-EDF dataset [24,66]	88.97 (Pz-Oz)
Seifpour et al. [77]	Sleep-EDF dataset [24,66]	90.60 (Fpz-Cz) 88.60 (Pz-Oz)
Sharma et al. [3]	Sleep-EDF dataset [24,66]	91.50 (Pz-Oz)
Zhou et al. [78]	1. Sleep-EDF database [24,66] 2. Sleep-EDF Expanded database [24]	1. 91.80 (Fpz-Cz) 2. 85.30 (Pz-Oz)
Zhang et al. [79]	1. UCD dataset [24] 2. MIT-BIH polysomnographic database [24]	1. 88.40 (C3-A2 + C4-A1) 2. 87.60 (C3-A2 + C4-A1)
Liu et al. [80]	Sleep-EDF Expanded database [24]	84.44 (Fpz-Cz + Pz-Oz)
Cai et al. method [81]	Sleep-EDF database [24,66]	87.21 (Fpz-Cz)
Loh et al. [82]	CAP Sleep Database [24,44]	90.46 (C4-A1/C3-A2)
Sharma et al. [49]	CAP Sleep Database [24,44]	85.10 (F4-C4 + C4-A1)
Dhok et al. [83]	CAP Sleep Database [24,44]	87.45 (C4-C1/C3-A2)
Sharma et al. [84]	CAP Sleep Database [24,44]	83.30 (C4-A1 + F4-C4)
The proposed method	CAP Sleep Database on PhysioNet [24]	Case1: 95.43 (F4-C4) Case2: 91.05 (F4-C4) Case3: 92.31 (F4-C4)

## Data Availability

The CAP Sleep Database has been downloaded from Physionet [48].

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
