# Peer review of "L-Tetrolet Pattern-Based Sleep Stage Classification Model Using Balanced EEG Datasets"

_diagnostics, 2022, doi:10.3390/diagnostics12102510_

Round 1

Reviewer 1 Report

1)      The studies in Table 1 that used the same database as Table 7 should be included in Table 7. Furthermore, to better understand the comparison between the accuracy of the proposed method and the previous works, the classification accuracy for all the stages should be reported. The author can add it in the supplementary material.

2)      Table 2, please specify the gender and age range for each disorder

3)      Page 5 first paragraph, provide all the available channels in the database and mention why F4-C4 is selected

4)      Page 5 step 1, describe the size of the pooling filter and the stride for absolute average pooling, average pooling, and absolute maximum pooling.

5)      Page 5, Step 4, describe with formula how decomposing will be done. In this step maximum pooling is considered as the decomposition level, while Fig. 1 considers maximum pooling and then decomposition.  

6)      Fig. 1, the outputs of the feature concatenation section in levels 1 to 4 are not connected to any other block. Please clarify

7)      The L letter (L-184 tetrolet) of the Tetris game was employed for pattern identification (cite to the reference)

8)         The following three pooling methods were used: average pooling, absolute 192 average pooling, absolute maximum pooling, and maximum pooling. Four not three

9)      Eq(4) j should be I,2,…Ln if the average is overpaying or i=1,3,5,…ln if not overlapping.

10)  Fig. 2, plot matrix mtr before applying L-tetrolet patterns.

11)  Eq (12) and (13), are they applying to P1 and P2 separately? Please clarify.

12)  Eqs(15) and (16) are the same. Furthermore, considering k=8, there are 128 values, not 256.

13)    What is the purpose of Eqs(17) and (18)

14)  Table 4, add the sensitivity and specificity to the table

15)  Fig. 4, indicates each confusion matrix belongs to which case.

16)  Discussion, the number of features and figure 5 should be discussed before figure 4.

17) What would be the performance if case 1 is used for training and case 2 is used for testing? 

Author Response

1)      The studies in Table 1 that used the same database as Table 7 should be included in Table 7. Furthermore, to better understand the comparison between the accuracy of the proposed method and the previous works, the classification accuracy for all the stages should be reported. The author can add it in the supplementary material.

Ans: Thank you for that suggestion. We have updated Table 1 and Table 7.

Table 1. Literature review on sleep stage detection

Studies

Method

Classifier

Dataset

Channels

The results (%)

Abbasi et al. [29]

Convolutional Neural Network

Ensemble

Collected data

Multiple channels

Sensitivity: 78.44

Specificity: 96.49

Accuracy: 94.27

Li et al. [30]

Multi-layer convolutional neural networks

Auxiliary

SHHS dataset [31]

C3-A2, C4-A1, EOG

Accuracy: 85.12

Zaidi and Farooq [32]

Fourier Synchrosqueezed Transform Features

Support vector machine

DREAMS dataset

Cz-A1

Accuracy: 82.60

Sors et al. [33]

Deep Convolutional Neural Network

Convolutional Neural Network

The Sleep Heart Health Study dataset [34]

C4-A1, C3-A2

Accuracy: 87.00

Goshtasbi et al. [35]

Convolutional Neural Network

Softmax

SHHS dataset [31]

C4-A1, C3-A2

Accuracy: 81.30

Kappa: 74.00

Shahbakhti et al. [36]

Nonlinear analysis

Linear discriminant analysis

DREAMS dataset [37]

Fp1, O1, and CZ or C3

Accuracy: 92.50

Sensitivity: 89.90

Specificity: 94.50

Zhao et al. [38]

SleepContextNet

Softmax

1. SHHS dataset [31]

2. CAP dataset [19,39]

C4-A1 and C3-A2

1. Accuracy: 86.40

Kappa: 81.00

2. Accuracy: 78.80

Kappa: 71.00

Eldele et al. [40]

Multi-Resolution Convolutional Neural Network, Adaptive Feature Recalibration

Softmax

SHHS dataset [31]

C4-A1

Accuracy: 84.20

Kappa: 78.00

Yang et al. [41]

One-Dimensional Convolutional Neural Network, Hidden Markov model

One-Dimensional Convolutional Neural Network, Hidden Markov model

DRM-SUB dataset [37]

Pz-Oz

Accuracy: 83.23

Kappa: 76.00

Table 7. The comparison results

Study

Dataset

Accuracy result (%)

Bajaj and Pachori [60]

Sleep-EDF dataset [19,61]

88.47 (Pz-Oz)

Hassan et al. [62]

Sleep-EDF database [19,61]

90.69 (Pz-Oz)

Jiang et al. [63]

1. Sleep-EDF database [19,61]

2. Sleep-EDF Expanded database [19]

89.40 (Fpz-Cz)

88.30 (Pz-Oz)

Kanwal et al. [64]

Sleep-EDF database [19,61]

93.00 (Pz-Oz, PFz-Cz, EOG)

Basha et al. [65]

Sleep-EDF database [19,61]

90.20 (PFz-Cz)

Jadhav et al. [66]

Sleep-EDF Expanded database [19]

85.07 (PFz-Cz)

82.92 (Pz-Oz)

Michielli et al. [67]

Sleep-EDF database [19,61]

90.80 (Pz-Oz)

Huang et al. [68]

Sleep-EDF Expanded database [19]

84.60 (Fpz-Cz)

82.30 (Pz-Oz)

Kim et al. [69]

CAP Sleep Database on PhysioNet [19]

73.60 (unspecified)

Shanin et al. [70]

Collected data

92.00 (C3-C4)

Karimzadeh et al. [71]

Sleep-EDF dataset [19,61]

88.97 (Pz-Oz)

Seifpour et al. [72]

Sleep-EDF dataset [19,61]

90.60 (Fpz-Cz)

88.60 (Pz-Oz)

Sharma et al. [3]

Sleep-EDF dataset [19,61]

91.50 (Pz-Oz)

Zhou et al. [73]

1. Sleep-EDF database [19,61]

2. Sleep-EDF Expanded database [19]

1. 91.80 (Fpz-Cz)

2. 85.30 (Pz-Oz)

Zhang et al. [74]

1. UCD dataset [19]

2. MIT-BIH polysomnographic database [19]

1. 88.40 (C3-A2+C4-A1)

2. 87.60 (C3-A2+C4-A1)

Liu et al. [75]

Sleep-EDF Expanded database [19]

84.44 (Fpz-Cz + Pz-Oz)

Cai et al. method [76]

Sleep-EDF database [19,61]

87.21 (Fpz-Cz)

Loh et al. [77]

CAP sleep database [19,39]

90.46 (C4-A1/C3-A2)

Sharma et al. [44]

CAP sleep database [19,39]

85.10 (F4-C4+C4-A1)

Dhok et al. [78]

CAP sleep database [19,39]

87.45 (C4-C1/C3-A2)

Sharma et al. [79]

CAP sleep database [19,39]

83.30 (C4-A1 + F4-C4)

Loh et al. [80]

CAP sleep database [19,39]

90.46 (C4-A1/C3-A2)

The proposed method

CAP Sleep Database on PhysioNet [19]

Case1: 95.43 (F4-C4)

Case2: 91.05 (F4-C4)

Case3: 92.31 (F4-C4)

2)      Table 2, please specify the gender and age range for each disorder

Ans: Thank you. Indeed, age and gender are important properties when it comes to sleep stage classification. We corrected Table 2 by including a column on Age: Min-Max (average).

Table 2. Neurological status and numbers of subjects

The neurological status

F

M

Age: Min-Max (average)

Number of patients

No pathology (controls/normal)

9

7

23-42 (32.18)

16

Nocturnal frontal lobe epilepsy (NFLE)

19

21

14-67 (30.27)

40

REM behavior disorder (RBD)

3

19

58-82 (70.72)

22

Periodic leg movements (PLM)

3

7

40-62 (55.10)

10

Insomnia

5

4

47-82 (60.88)

9

Narcolepsy

3

2

18-44 (31.60)

5

Sleep-disordered breathing (SDB)

-

4

65-78(71.25)

4

Bruxism

-

2

23-34 (28.50)

2

Total number of pathologies

33

59

14-82 (49.19)

92

F: Female, M: Male

3)      Page 5 first paragraph, provide all the available channels in the database and mention why F4-C4 is selected

Ans: Dear reviewer, thank you for your valuable comment. We explained it as below.

These channels are the commonly used EEG channels [38,44,45]. Therefore, we used these channels.

4)      Page 5 step 1, describe the size of the pooling filter and the stride for absolute average pooling, average pooling, and absolute maximum pooling.

Ans: Dear reviewer, thank you for that valuable comment. We clarified this issue with the statement below.

Herein, we used non-overlapping blocks with a length of two to create decompressed signals. In the M1 function, the absolute value of the used non-overlapping block was used as the decompressed signal. In M2 and M3, we used the maximum of the absolute block values and average values of the used non-overlapping blocks for decomposition. Eqs. 3-8 provide a mathematical definition of these functions.

5)      Page 5, Step 4, describe with formula how decomposing will be done. In this step maximum pooling is considered as the decomposition level, while Fig. 1 considers maximum pooling and then decomposition.  

Ans: Dear reviewer, thank you for your valuable comment. We gave formulas of the used maximum pooling as below.

The used maximum pooling operator has been given in Equation 1.

(1)

(2)

Herein,  defines the maximum pooling function,  is decomposed signal,  is the length of the used EEG signal (), and  is maximum value finding function.

6)      Fig. 1, the outputs of the feature concatenation section in levels 1 to 4 are not connected to any other block. Please clarify

Ans: Thank you for your valuable comment. We the edited the figure and added the missing connections.

7)      The L letter (L-184 tetrolet) of the Tetris game was employed for pattern identification (cite to the reference)

Ans: The relevant text is cited in the revised version of the manuscript.

The letter `L’ (L-tetrolet) of the Tetris game was employed for pattern identification [48,49].

8)         The following three pooling methods were used: average pooling, absolute 192 average pooling, absolute maximum pooling, and maximum pooling. Four not three

Ans: This point was corrected as follows:

By deploying four pooling functions – absolute average pooling, absolute maximum pooling, average pooling, and maximum pooling –, a five leveled feature generation method was created.

9)      Eq(4) j should be I,2,…Ln if the average is overpaying or i=1,3,5,…ln if not overlapping.

Ans: We corrected it as below.

The used maximum pooling operator has been given in Equation 1.

(1)

(2)

Herein,  defines the maximum pooling function,  is decomposed signal,  is the length of the used EEG signal (), and  is maximum value finding function.

10)  Fig. 2, plot matrix mtr before applying L-tetrolet patterns.

Ans: We have clarified that point with the drawing below:

Figure 2 depicts the resulting 4 x 4 matrix.

Fig. 2. The created 4 x 4 matrix for applying the proposed L-tetrolet pattern.

11)  Eq (12) and (13), are they applying to P1 and P2 separately? Please clarify.

Ans: Dear reviewer, thank you for your incredible attention. We clarify this issue with the figures and text below.

Fig. 3. The used L-tetrolet patterns. Each L-tetrolet is named using a letter (e.g., a, b, c, d), and these letters are shown using different colors. These patterns are called P1 and P2

Step 2.4: Extract bits using P1, P2, and binary feature generation function . These patterns (P1 and P2) are separately applied to the generated matrix.

(14)

(15)

(16)

where  and  are the first and second parameters of the binary feature generation (signum) function. Eqs. 11-13 were deployed to both P1 and P2, and eight bits were extracted from each pattern. The extracted bits are named  and . By deploying these bits, two novel signals were created for feature generation, and these signals were named the first map () and the second map signal (), respectively. Binary to decimal conversion was used to create these signals, as shown in Eqs. 17-18.

Step 2.5: Create map signals employing the generated bits.

(17)

(18)

12)  Eqs(15) and (16) are the same. Furthermore, considering k=8, there are 128 values, not 256.

Ans: We corrected them.

Step 2.5: Create map signals employing the generated bits.

(17)

(18)

13)    What is the purpose of Eqs(17) and (18)

Ans:  Dear reviewer. We corrected it per your valuable comment.

(19)

(20)

where  and  are histograms of the first and second map signals respectively.  function is defined to extract histogram.

14)  Table 4, add the sensitivity and specificity to the table

Ans: Thanks for your valuable comment. We added these parameters.

Sensitivity

Specificity

90.27

98.94

86.22

97.17

87.03

97.96

15)  Fig. 4, indicates each confusion matrix belongs to which case.

Ans: Thank you for your constructive comments. We corrected this figure.

16)  Discussion, the number of features and figure 5 should be discussed before figure 4.

Ans: Thank you for your valuable comments. We reorganized the discussion section as below.

As stated in Sect. 3, the presented model has three fundamental phases: feature generation, TSRFINCA-based feature selection, and classification. The presented model uses four pooling methods to overcome the routing problem of the pooling. For instance, the maximum pooling only routes peak values. We proposed a multiple pooling-based decomposition model to overcome the routing problem. Both textural and statistical feature generators have been utilized for feature extraction to create handcrafted features. By using these feature extractors and the proposed multiple pooling function, a multileveled feature extraction method has been presented to generated features at both low and high level. The presented three-layered feature selection function – TSRFINCA – selected the top informative features from the used datasets. In this research, we have used three datasets. The presented TSRFINCA selected a variable-sized feature vector for each dataset. The sizes of the optimal feature sets were found as 644, 711, and 188 for Case 1, Case 2, and Case 3, respectively. Figure 5 documents the feature selection process.

Fig. 5. Feature selection processes of the cases.

In this figure (see Figure 5), number of features and loss values are demonstrated. The proposed feature selector is an iterative feature selector and it calculates loss values of the 901 (initial value and end value of the loop are 100 and 1000 respectively. Thus, 901=1000-100+1 feature vectors have been evaluated for each dataset) feature vectors. The optimal feature vectors have been selected using minimum loss values. These optimal features were forwarded to the CSVM classifier. This classifier was utilized as both a loss value generator (calculate misclassification rates of the chosen 901 feature vectors) and a classifier. Figure 6 shows a confusion matrix for each case.

Fig. 6. The calculated confusion matrices of each case

The confusion matrices, in Figure 6 denote the case specific results.

17) What would be the performance if case 1 is used for training and case 2 is used for testing?

Ans: Dear reviewer, we calculated it and it is about 50%.

Reviewer 2 Report

It has been a comprehensive study, and I think it has valuable content. The subject is fascinating. However, the following significant corrections seem necessary to improve the scientific level of the article.

1 - The reviewer doubts that these results can be generalized. Please implement the proposed method on another open-access dataset.

2- The "Introduction" section needs a major revision to provide a more accurate and informative literature review of the pros and cons of the available approaches and how the proposed method is different comparatively. Also, the motivation and contribution should be stated more clearly.

3- The logic of the introduction can be improved. For example, the reasons and significance of applying machine methods to detect sleep stages. Current progress and critical issues could also be mentioned. The authors can use these articles to edit this section. An improved neural network based on SENet for sleep stage classification, Deep Convolutional Recurrent Model for Automatic Scoring Sleep Stages Based on Single-Lead ECG Signal, Automated Characterization of Cyclic Alternating Pattern Using Wavelet-Based Features and Ensemble Learning Techniques with EEG Signals, Automated detection of driver fatigue from electroencephalography through wavelet-based connectivity, EEG Sub-bands based Sleep Stages Classification using Fourier Synchro squeezed Transform Features

4- The conclusion should be more carefully rewritten, summarizing what has been learned and why it is interesting and valuable.

5- What about the computational complexity (e.g., running time analysis)?

Author Response

It has been a comprehensive study, and I think it has valuable content. The subject is fascinating. However, the following significant corrections seem necessary to improve the scientific level of the article.

1 - The reviewer doubts that these results can be generalized. Please implement the proposed method on another open-access dataset.

Ans: Dear reviewer, thank you for your valuable comment. We tested this model already on the three different datasets and we gave the results for each case. 

2- The "Introduction" section needs a major revision to provide a more accurate and informative literature review of the pros and cons of the available approaches and how the proposed method is different comparatively. Also, the motivation and contribution should be stated more clearly.

Ans: We corrected literature review as per your valuable comment.

Table 1. Literature review on sleep stage detection

Studies

Method

Classifier

Dataset

Channels

The results (%)

Abbasi et al. [29]

Convolutional Neural Network

Ensemble

Collected data

Multiple channels

Sensitivity: 78.44

Specificity: 96.49

Accuracy: 94.27

Li et al. [30]

Multi-layer convolutional neural networks

Auxiliary

SHHS dataset [31]

C3-A2, C4-A1, EOG

Accuracy: 85.12

Zaidi and Farooq [32]

Fourier Synchrosqueezed Transform Features

Support vector machine

DREAMS dataset

Cz-A1

Accuracy: 82.60

Sors et al. [33]

Deep Convolutional Neural Network

Convolutional Neural Network

The Sleep Heart Health Study dataset [34]

C4-A1, C3-A2

Accuracy: 87.00

Goshtasbi et al. [35]

Convolutional Neural Network

Softmax

SHHS dataset [31]

C4-A1, C3-A2

Accuracy: 81.30

Kappa: 74.00

Shahbakhti et al. [36]

Nonlinear analysis

Linear discriminant analysis

DREAMS dataset [37]

Fp1, O1, and CZ or C3

Accuracy: 92.50

Sensitivity: 89.90

Specificity: 94.50

Zhao et al. [38]

SleepContextNet

Softmax

1. SHHS dataset [31]

2. CAP dataset [19,39]

C4-A1 and C3-A2

1. Accuracy: 86.40

Kappa: 81.00

2. Accuracy: 78.80

Kappa: 71.00

Eldele et al. [40]

Multi-Resolution Convolutional Neural Network, Adaptive Feature Recalibration

Softmax

SHHS dataset [31]

C4-A1

Accuracy: 84.20

Kappa: 78.00

Yang et al. [41]

One-Dimensional Convolutional Neural Network, Hidden Markov model

One-Dimensional Convolutional Neural Network, Hidden Markov model

DRM-SUB dataset [37]

Pz-Oz

Accuracy: 83.23

Kappa: 76.00

We revised motivation and contributions sides as below.

Our main motivations were to propose a game-based feature extraction function and, by applying this function, present a new EEG signal classification model. To achieve that highly accurate learning model, a new L-tetrolet pattern and TSRFINCA based sleep EEG signal classification model was created. The L-tetrolet pattern for textural feature extraction was inspired by the Tetris game. Statistical features were also extracted to enforce the presented feature generation method. A multilevel feature generation architecture was created using pooling functions to generate low-level and high-level features. The presented feature selector (TSRFINCA) incorporated three stages. In the first stage, a threshold point is determined, and feature selection is carried out by deploying this threshold point. ReliefF is applied to the selected features in the second stage, and the positive weighted feature is selected. In the last stage, Iterative Neighborhood Component Analysis (INCA) is applied to the selected features, and the most meaningful features are selected. The selected final features are utilized as the input of the Cubic Support Vector Machine (CSVM) classifier. To summarize, we proposed (i) a new game-based feature extractor, (ii) a new decomposition model by using four pooling techniques, and (iii) a hybrid high-performed feature selector. These methods have been used in a feature engineering model to get high classification performance.

Contributions:

  • A new feature engineering model has been created by proposing new generation feature extraction, decomposition, and feature selection methods. The essential purpose of the proposed feature engineering model is to extract the most informative features from the used signals to get high classification performance with low time complexity.
  • This research presents a highly accurate EEG classification model for sleep stage detection. By deploying the presented classification model, sleep stage classification results of the CAP sleep dataset are presented using three cases. Our proposal denotes general high classification performance since we applied this model to three different datasets.

3- The logic of the introduction can be improved. For example, the reasons and significance of applying machine methods to detect sleep stages. Current progress and critical issues could also be mentioned. The authors can use these articles to edit this section. An improved neural network based on SENet for sleep stage classification, Deep Convolutional Recurrent Model for Automatic Scoring Sleep Stages Based on Single-Lead ECG Signal, Automated Characterization of Cyclic Alternating Pattern Using Wavelet-Based Features and Ensemble Learning Techniques with EEG Signals, Automated detection of driver fatigue from electroencephalography through wavelet-based connectivity, EEG Sub-bands based Sleep Stages Classification using Fourier Synchro squeezed Transform Features

Ans: We corrected literature review per your valuable comment. The relevant papers were cited.

Table 1. Literature review on sleep stage detection

Studies

Method

Classifier

Dataset

Channels

The results (%)

Abbasi et al. [29]

Convolutional Neural Network

Ensemble

Collected data

Multiple channels

Sensitivity: 78.44

Specificity: 96.49

Accuracy: 94.27

Li et al. [30]

Multi-layer convolutional neural networks

Auxiliary

SHHS dataset [31]

C3-A2, C4-A1, EOG

Accuracy: 85.12

Zaidi and Farooq [32]

Fourier Synchrosqueezed Transform Features

Support vector machine

DREAMS dataset

Cz-A1

Accuracy: 82.60

Sors et al. [33]

Deep Convolutional Neural Network

Convolutional Neural Network

The Sleep Heart Health Study dataset [34]

C4-A1, C3-A2

Accuracy: 87.00

Goshtasbi et al. [35]

Convolutional Neural Network

Softmax

SHHS dataset [31]

C4-A1, C3-A2

Accuracy: 81.30

Kappa: 74.00

Shahbakhti et al. [36]

Nonlinear analysis

Linear discriminant analysis

DREAMS dataset [37]

Fp1, O1, and CZ or C3

Accuracy: 92.50

Sensitivity: 89.90

Specificity: 94.50

Zhao et al. [38]

SleepContextNet

Softmax

1. SHHS dataset [31]

2. CAP dataset [19,39]

C4-A1 and C3-A2

1. Accuracy: 86.40

Kappa: 81.00

2. Accuracy: 78.80

Kappa: 71.00

Eldele et al. [40]

Multi-Resolution Convolutional Neural Network, Adaptive Feature Recalibration

Softmax

SHHS dataset [31]

C4-A1

Accuracy: 84.20

Kappa: 78.00

Yang et al. [41]

One-Dimensional Convolutional Neural Network, Hidden Markov model

One-Dimensional Convolutional Neural Network, Hidden Markov model

DRM-SUB dataset [37]

Pz-Oz

Accuracy: 83.23

Kappa: 76.00

4- The conclusion should be more carefully rewritten, summarizing what has been learned and why it is interesting and valuable.

Ans: Dear reviewer. Thanks for your valuable comment. We corrected this section as below.

Conclusions

In this research, we propose a new feature engineering model. The essential goal of that model is to extract the most significant features from EEG signals. The model is based on a new game-based feature extraction function, named L-tetrolet which extracts textural feature information. To generate high level features, a multileveled feature extraction structure is presented using a combination of four pooling techniques. This approach fuses hybrid approximation and the advantages of pooling techniques. In the feature selection phase, a three-layered hybrid feature selector has been used, and the selected features have been classified using a shallow classifier. Using PhysioNet, three different sleep EEG datasets were created, containing six groups. Our proposed L-tetrolet-based model attained >90% overall classification accuracy on these datasets. Moreover, our proposal reached 95.43% classification accuracy in Case 1. These results were compared to other recent models, showing that our model outperforms all the previous methods used for sleep stage detection based on signals from the CAP database. These findings demonstrated that our model achieved satisfactory classification performance and time complexity for solving sleep stage classification problems using EEG signals.

In the future, we plan to:

  • Propose new game-based feature extraction functions.
  • Purpose self-organized feature engineering models.
  • Propose a new generation of pooling/decomposition methods by using quantum computing and superposition.
  • Develop a new sleep stage classification application which will be used in medical centers.

5- What about the computational complexity (e.g., running time analysis)?

Ans: Dear reviewer, thank you for your valuable comment. We added the following text to our manuscript.

Computational complexity is a crucial property that determines the practicality of the proposed model. A lower computational complexity is more resource-efficient which translates into less energy usage and lower cost. The presented model consists three of algorithms. Therefore, the time complexities of these algorithms should be calculated [52,53]. Table 6 introduces these calculations in detail.

Table 6. The time complexity calculation of the presented model.

Phase

Steps

Computational complexity

Feature generation

Pooling based decomposition

Statistical feature generation

Textural feature generation (L-tetrolet pattern)

Statistical features extraction of the textural features

TSRFINCA

Threshold feature selection

ReliefF based selection

INCA

Classification

SVM

Total

Round 2

Reviewer 1 Report

The authors addressed most of my comments. However, there are still some issues to be addressed: 

1)In M2 and M3, we used the maximum of the absolute block values and the average values of the used non-overlapping blocks for decomposition should be changed to 

In M2 and M3, we used the  average values and maximum of the absolute block values of the used non-overlapping blocks for decomposition 

2) In Fig. 2 the matrix elements should be based on a, b, c, and d.

3) Eqs. 11-13 were deployed to both P1 and P2, should be Eqs: 13-15

4) The name of electrodes used in each database (Case 1 and Case 2) should be mentioned in the text. 

5) The authors should mention in the paper the performance for the case when they train the algorithm with Case 1 and test with case 2 and should modify the features to increase the generalizability of the algorithm to get a good result for this case or discuss why their performance is dropped considerably by training with case 1 and testing with case 2. 

Reviewer 2 Report

The manuscript was modified very well. The authors have attempted to address all of the reviewers' comments in the revised paper. The manuscript seems acceptable to me for publication in the journal with the corrections made. 

Round 3

Reviewer 1 Report

The authors addressed all my comments. Please correct the following two typos:

Eq. 14 right-hand side column 1 row 4, V13 should be changed to V10. 

Change "The datasets used are small. Therefore, when we used one dataset for training and the other datasets for training" to "The datasets used are small. Therefore, when we used one dataset for training and the other datasets for testing"
